# Response to Omalizumab as an Add-On Therapy in the Treatment of Allergic Asthma in Adult Chinese Patients—A Retrospective Study

**DOI:** 10.3390/vaccines10122068

**Published:** 2022-12-02

**Authors:** Na Li, Linfeng Cao, Ming Zhang, Chunyuan Fei, Jingjing Deng

**Affiliations:** Department of Pulmonary and Critical Care Medicine, The First Hospital of Jiaxing, Affiliated Hospital of Jiaxing University, Jiaxing 314001, China

**Keywords:** omalizumab, budesonide formoterol, immunoglobulin E, allergic asthma, response

## Abstract

(a) Background: Omalizumab is an anti-IgE humanized monoclonal antibody marketed in China for the conventional treatment of poorly controlled moderate-to-severe allergic asthma. Numerous clinical trials have demonstrated the effectiveness of omalizumab, but the data from studies in actual clinical treatment are still relatively limited. (b) Methods: Thirty-two patients with moderate-to-severe allergic asthma treated with omalizumab on the basis of ICS-LABA (inhaled corticosteroids/long-acting beta2-agonist) were selected. Clinical characteristics before and after treatment were collected to analyze the relationship between changes in serum total IgE levels and peripheral blood EOS (eosinophil) levels, FEV1 (forced expiratory volume in 1 second), PEF (peak expiratory flow), OCS (oral glucocorticoid) dosage, ATC (asthma control test) score, and the number of acute exacerbations and the treatment response, in order to observe the efficacy of omalizumab in addition to primary therapy, and to investigate whether baseline clinical characteristics such as serum total IgE and EOS levels could predict a treatment response. (c) Results: Using the ACT score as an evaluation, 68.75% of patients benefited from omalizumab treatment at the end of 16 weeks. The response group has a reduction in OCS dosage (*p*-values of 0.026 and 0.039), a significant reduction in ACT scores (both *p* < 0.001), and a reduction in the number of acute exacerbations (*p* = 0.034 and 0.025, respectively) after omalizumab treatment. The binary logistics analysis of factors affecting the effectiveness of omalizumab in the treatment of allergic asthma were total serum IgE and the presence of comorbidities (*p*-values of 0.039 and 0.046, respectively). (d) Conclusions: Combining omalizumab with ICS-LABA for 16 weeks significantly improves asthma symptoms in Chinese adults and can be used as an add-on treatment. In addition, high serum IgE levels and the presence of comorbidities were predictors of its therapeutic efficacy.

## 1. Introduction

Asthma is a chronic inflammatory disease with both airway hyperresponsiveness and variable airflow limitation, characterized by recurrent episodes of wheezing and shortness of breath, with or without chest tightness or cough [1]. It has different clinical phenotypes [2], with allergic asthma being the most widespread and easily identifiable phenotype, accounting for 60–80% of cases. In 2015, 358 million people were reported to have asthma worldwide, an increase of 12.6% in prevalence from 1990 [3]; in China, in the same year, 45.7 million people over the age of 20 suffered from asthma, extrapolated from the 2015 national census [4]. Even though the concept of overall asthma control has been promoted for many years [5], the current state of control is still not satisfactory—although there has been progress—with asthma control rates of around 28.5% in urban areas [6], and these may be even lower in remote areas and primary hospitals. Timely and effective asthma control and management is the goal of asthma control, and this is pursued by medical professionals worldwide.

Immunoglobulin E [7] is an important factor in the persistence and exacerbation of allergic asthma symptoms, and its mediated type I hypersensitivity reaction plays a role in the pathophysiological mechanisms of the allergic response, as well as in the inflammation of the asthmatic airways. Based on the pathological mechanism of IgE in allergic asthma and the blocking of the IgE-mediated immune pathway, omalizumab (OMA) was approved as an anti-IgE humanized monoclonal antibody for the treatment of patients with poorly controlled moderate-to-severe asthma, and was the first targeted drug used in the field of asthma treatment [8]. The clinical data [9,10,11] and experience accumulated since its approval by the USFDA (United States Food and Drug Administration) in 2003 have shown that omalizumab can reduce acute asthma exacerbations, improve symptoms, enhance life treatment, and reduce systemic glucocorticoid doses.

Glucocorticoids are the cornerstone of asthma treatment and play an irreplaceable role. Inhaled corticosteroids (ICS) have become the drug of choice for long-term treatment because of their effective local anti-inflammatory effects and low adverse effects. According to the 2022 Global Initiative for Asthma (GINA) [1] guidelines for a five-stage regimen for the long-term treatment of asthma, good asthma control can be achieved with ICS alone in patients with mild asthma. The majority of patients with moderate-to-severe bronchial asthma tend to require combination therapy, with common combinations of leukotriene modulators, long-acting beta2-agonist (LABA), extended-release theophylline, methanesulfonate, and sodium cromoglycate. Budesonide/formoterol powder inhalation is a common clinical combination in China. It is administered twice daily via inhalation through a simplified treatment dry powder inhaler (DPI), providing both bronchodilator and anti-inflammatory activity, with a therapeutic effect that is equivalent to or even better than doubling the efficacy of ICS. It also reduces the side-effects of high-dose ICS, and it plays a pivotal role in the clinical management of patients with moderate-to-severe asthma [12]. Although the combination of ICS-LABA has now shown good clinical efficacy as the first choice for asthma control, there are still some patients with poor control and decreased sensitivity. Omalizumab as an add-on therapy has demonstrated the benefit of targeted IgE therapy in the treatment of allergic asthma and other allergic diseases.

Omalizumab has more than 10 years of clinical experience abroad, and its efficacy and safety have been proven. However, differences in internal and external factors such as different ethnic groups, geography, and dietary structure, especially for omalizumab as a biologic agent, may significantly exacerbate these differences. Omalizumab was launched in mainland China in August 2017, and thus far, it is still in its infancy, with clinical application experience, effects, and relevant data in the domestic population still lacking. In order to enrich the clinical application data and to better guide its clinical use, this study compared the changes in EOS (eosinophil) count and EOS percentage, total serum IgE level, FEV1 (forced expiratory volume in 1 second), PEF (peak expiratory flow), ACT (asthma control test) score, allergic status, and the presence of comorbidities before and after 16 weeks of omalizumab treatment in allergic asthma adult patients on the basis of budesonide/formoterol powder inhalation, to observe the efficacy of omalizumab in combination with budesonide/formoterol in the treatment of moderate-to-severe allergic asthma, and to investigate whether baseline clinical characteristics such as total serum IgE and EOS levels are predictive of a response to treatment.

## 2. Materials and Methods

This study protocol was approved by the institutional ethics committee, and all subjects gave informed consent.

### 2.1. Clinical Cases Data

Thirty-two patients with moderate-to-severe allergic asthma treated with omalizumab in the Department of Pulmonary and Critical Care Medicine of the First Hospital of Jiaxing from October 2020 to June 2022 were enrolled. 

Inclusion criteria: (a) moderate-to-severe asthma patients aged ≥ 12 years who met the criteria of the Asthma Group of the Chinese Thoracic Society [13] (*Guidelines for bronchial asthma prevention and management*, 2020 edition)-moderate asthma was defined as those who could achieve complete control using grade 3 therapy, and severe asthma was defined as fully or incompletely controlled with grade 4 or 5 asthma medications; (b) confirmed allergy status: elevated total serum IgE, positive specific IgE test, positive skin prick test, or a combination of other allergic diseases (e.g., allergic rhinitis, atopic dermatitis, food allergy, etc.); (c) poor control after ≥3 months of conventional 320/9 mcg twice daily budesonide/formoterol therapy; (d) treatment with omalizumab.

Exclusion criteria: (a) hypersensitivity to budesonide/formoterol DPI or to the active ingredient omalizumab; (b) receiving allergen-specific immunotherapy (AIT) or other biologically targeted therapies (e.g., anti-IL-5 monoclonal antibody, anti-IL-4 monoclonal antibody, anti-IL-13 monoclonal antibody, anti-IL-5Ra monoclonal antibody, etc.); (c) acute asthma exacerbations; and (d) combined with diseases that severely affect ventilation, such as bronchiectasis, lung cancer, allergic bronchopulmonary aspergillosis (ABPA), acute respiratory infections, etc.

### 2.2. Methods and Clinical Data

Clinical data were collected retrospectively from the hospital’s electronic medical record system from patients who met the inclusion criteria and who had received budesonide/formoterol in combination with omalizumab, before and after 16 weeks of treatment. The baseline data collected before patients starting omalizumab treatment included: demographic characteristics such as age, sex, weight, and body mass index (BMI); course of asthma, co-morbid allergic diseases, comorbidities, blood count, total serum IgE, FEV1, PEF, oral glucocorticoid (OCS) dose, asthma control test (ACT) score, number of acute exacerbations, and other clinical characteristics. The changes before and after 16 weeks of treatment were recorded, collected, and documented.

Patients were assessed for asthma control before and after treatment using the ACT score recommended by the Chinese guidelines [13,14], and were divided into non-response and response groups. The total ACT score ranged from 5 to 25. An ACT score of 20–25 indicated good asthma control, 16–19 indicated poor asthma control, and 5–15 indicated very poor asthma control. The minimum clinically significant change (MID) in ACT was 3 [15,16].

In this study, a response after 16 weeks of omalizumab treatment was required to meet any of the following conditions: (a) an improvement in ACT score ≥ 3 (MID); and (b) a pre-treatment ACT score < 20 (poor or poorly controlled asthma) and a post-treatment ACT score ≥ 20 (well controlled asthma). In pulmonary function tests: the FEV1 unit is liters per second (L/s); PEF is in liters (L). ICS doses were converted to equivalent budesonide doses (mg/d), and OCS doses were converted to equivalent prednisone doses (mg/d). The normal range of reference values for serum total IgE antibodies is 0–100 ng/mL. A clear history of allergy is defined as a clear occurrence of an allergic reaction to inhalation, ingestion, and contact with certain items or the use of certain drugs, Inhaled allergens include tree assemblages (willow/poplar/elm), common ragweed, Artemisia, house dust, cat hair, dog epithelium, cockroaches, mold assemblages (*Penicillium punctatum*/branch molds/*Trichoderma* spp.), Humulus, and cross-reactive sugar antigen determinants, while ingested allergens include egg whites, milk, peanuts, sea fish assemblages (cod/lobster/scallop), soy, beef, lamb, shrimp, and crab.

### 2.3. Statistical Methods

SPSS software version 26.0 (SPSS Inc., Cary, NC, USA) and GraphPad Prism 7 software (GraphPad Software, Inc., San Diego, CA, USA) were applied for statistical analysis and graph production. Continuous variables that conformed to a normal distribution were expressed as mean ± standard deviation (SD); those that did not conform to a normal distribution were expressed as a median (interquartile range, IQR); categorical variables were expressed as percentages. Comparisons between groups of continuous variables were evaluated using a *t*-test (conforming to a normal distribution) or a Mann-Whitney U-test (not conforming to a normal distribution), and a chi-squared test for categorical variables. Differences in the baseline data and changes in clinical parameters before and after treatment were compared between non-response and response outcomes, respectively. Paired-sample *t*-tests were used for compliance with a normal distribution, and Wilcoxon tests were used for paired and non-normally distributed data. To analyze the predictors of treatment response after 16 weeks, and to compare the differences in clinical parameters between the non-response and response outcomes, binary logistic regression analysis was used to analyze the relationship between total serum IgE, the presence of comorbidities, EOS count, whether serum-specific IgE was positive, and FEV1 and PEF response or non-response, to derive odds ratios (OR) and 95% confidence intervals (CI); and Cox regression forest plots were produced. Values of *p* < 0.05 were considered to represent a statistically significant difference, and all *p*-values were the result of two-sided tests.

## 3. Results

### 3.1. Baseline and Clinical Characteristics

A total of 32 patients with moderate-to-severe allergic asthma treated with subcutaneous omalizumab were included. ICS-LABA contained both budesonide formoterol 320ug/9ug/suction from AstraZeneca, and omalizumab from Novartis Pharma Ltd. All patients were treated with budesonide formoterol 320 plus omalizumab via subcutaneous injection. The dose and dosing interval of omalizumab were determined according to the patient’s total serum IgE level (ng/mL = 2 IU/mL) and body weight (kg) prior to treatment, and according to the Chinese dosing schedule recommended in the drug instructions. Doses of omalizumab 150–600 mg were given subcutaneously every 4 weeks (Table 1). Drug instructions: https:/www.xolairhcp.com/starting-treatment/dosing.html, accessed on 1 December 2022

A total of 13 male patients (40.63%) and 19 female patients (59.37%) with an asthma course of 4.00 (2.00–10.00) years were included. Gender, age, BMI, and the course of disease were comparable in on-responders and responders, *p* > 0.05. There were two cases of combined rhinosinusitis (RS, 6.25%), six cases of asthma–COPD overlap (ACO 18.75%), and one case of gastro-esophageal reflux disease (GERD, 3.13%). Among the allergy states, three cases (9.38%) were allergic to food, six cases (18.75%) were allergic to inhalants, seven cases (21.87%) were allergic to drugs, three cases (9.38%) were allergic to skin (chronic urticaria and atopic dermatitis), of which totaled seven cases (21.87%) of patients with allergic rhinitis, and three patients had more than six allergens in serum-specific IgE detection. The median total serum IgE level was 503.80 (315.68–956.00) ng/ml, and the median peripheral blood EOS count and percentage were 0.43 × 10^9^/L (0.10−0.78 × 10^9^/L) and 5.20% (1.63–8.40%), respectively; the mean values of FEV1 and PEF were 2.06 ± 0.63 (L/s) and 4.56 ± 0.90 (L). Patients had a poor level of control at baseline with a mean ACT score of 16.38 ± 1.91 and a median number of episodes of 0.00 (0.00–1.00)/3 months, with two of the non-responders having 4 episodes/3 months. Patients were all on 320 budesonide formoterol DPI prior to initial treatment, and 11 (34.38%) were also additionally on OCS at a median dose of 0.00 (0.00–13.75). The differences in baseline characteristics between non-response and response were statistically significant between the OCS dose, total serum IgE, and the number of episodes, with *p*-values of 0.024, 0.016, and <0.001, respectively (Table 2).

### 3.2. Efficacy Assessment after 16 Weeks

Before and after 16 weeks of omalizumab treatment, OCS dosage, total serum IgE, EOS count and percentage, FEV1, and the number of acute exacerbations decreased, and the ACT score improved, all with statistically significant differences (*p* < 0.05), suggesting that the overall efficacy of omalizumab was good, reflecting a reduced OCS dosage, decreased number of acute exacerbations, and improved asthma control after treatment (Table 3).

Further, we analyzed the differences between the non-response and response groups, and the statistical results of the paired data showed a reduction in OCS dosage (*p*-values of 0.026 and 0.039), a significant improvement in ACT scores (both *p* < 0.001), and a reduction in the number of acute exacerbations (*p* = 0.034 and 0.025, respectively) after omalizumab treatment; irrespective of whether the patient met the response criteria, they still had improved symptoms and a reduced oral hormone use. Of course, the responding group had more significant serum total IgE, EOS counts and percentages, and FEV1, which could explain more fully the evaluation of the efficacy of omalizumab in allergic asthma (Table 4, Figure 1).

### 3.3. Predicting the Response to Omalizumab

A binary logistic regression analysis using the total serum IgE, the presence of comorbidities (RS, ACO, and GERD), EOS count, whether the serum IgE was positive, and FEV1 and PEF as correlates yielded an R^2^ = 0.242; the OR, 95CI, and *p*-values are shown in the Cox forest plot (Figure 2). Based on the results, it can be tentatively concluded that the factors influencing the effectiveness of omalizumab in the treatment of allergic asthma are total serum IgE and the presence of comorbidity (*p*-values of 0.039 and 0.046, respectively), with patients with a high total serum IgE and comorbidity being less likely to respond and potentially showing a lower effectiveness in treatment.

## 4. Discussion

This study investigated the therapeutic effect of omalizumab in a population with moderate-to-severe allergic asthma in a third-tier city (small city size) in China. It can improve overall asthma control, reduce the dosage of OCS, and reduce the number of acute exacerbations, similar to the results of the few other studies in first-tier cities in China. Using the ACT score as an evaluation, 68.75% of patients benefited from omalizumab treatment at the end of 16 weeks; high total serum IgE and patients with comorbidities predicted a poorer response to omalizumab treatment, and high eosinophil counts or percentage did not predict the response to omalizumab treatment. All of the patients in this study were treated with ICS-LABA as a basic therapy, with some patients being additionally treated with a controller drug such as OCS, but most patients still had poor symptom control and baseline ACT scores of below 20, reflecting the heavy disease burden in asthma patients and highlighting the need to initiate omalizumab as an additional therapy in these patients.

Several of the main biologically targeted drugs currently available for asthma target key pathways in the pathogenesis of asthma are: IgE, interleukin (IL)-5, and IL-4/IL-13, all of which modulate Th2-type inflammation, and which therefore may share some common characteristics in their target populations [13]. However, not all asthma patients will benefit from biologically targeted therapies, and they are currently expensive. Therefore, the selection of asthma populations that are more likely to benefit from biologically targeted drugs based on appropriate biomarkers and clinical characteristics, as well as guiding the continuation or discontinuation of targeted drugs, is an important clinical issue. The existing asthma biological targeting agents primarily target two key players in Th2 inflammation: IgE (omalizumab) and eosinophils (mepolizumab, reslizumab, and benralizumab). It is suggested that the biomarkers of Th2 inflammation such as IgE, peripheral blood, or induced sputum eosinophils, FeNO, and osteochondral proteins may be used to guide biologically targeted therapies for asthma [17,18]. Mepolizumab, reslizumab, and benralizumab bind to IL-5 or IL-5 receptors, and act on eosinophils. Studies have shown that patients with peripheral blood eosinophilia (≥150/μL) benefit more from anti-IL-5 therapy, and it is therefore recommended for severe asthma with marked eosinophilia, regardless of whether it is allergic or not. However, unlike eosinophil levels that reliably predict a response to anti-IL-5 therapy, it remains controversial as to whether high eosinophil levels can predict a response to treatment despite the mechanism of action of anti-IgE involving Th2-type inflammation. Similarly, the use of IgE levels to predict a response to omalizumab therapy is controversial, although the total serum IgE levels and body weight are recommended to determine the dose of omalizumab. 

Total serum IgE includes both active free IgE and IgE that has been inactivated via binding to omalizumab; after starting omalizumab treatment, the total serum IgE level increases because the half-life of IgE is increased by the binding of omalizumab to IgE. The total IgE levels at this time do not reflect free IgE levels, and therefore it is not recommended to evaluate the effect of treatment based on the changes in total serum IgE levels after treatment [19,20]. Studies [21,22] to measure the diagnostic predictive value of free IgE levels for the efficacy of omalizumab are therefore ongoing, but they are currently sparse, and the results suggest that they cannot be used to predict or to assess treatment efficacy.

However, a number of studies [23,24] suggest that the administration of omalizumab therapy using total serum IgE levels is a reliable approach. GINA [1] and our guidelines [13] similarly recommend a serum total IgE level in the range of 30–1500 IU/mL as an indication for the use of omalizumab, and they do not recommend its use in patients with a total IgE of greater than 1500 IU/mL, due to the lack of evidence-based support. In clinical practice, however, a significant number of patients with moderate-to-severe allergic asthma have been found to have total serum IgE levels of above 1500 IU/mL. More clinical studies are needed to evaluate the use, benefits, and safety of omalizumab in this group of asthmatics, and to expand the potential beneficiary population of omalizumab. Eosinophils are one of the major inflammatory cells involved in the inflammatory response in bronchial asthma and other allergic diseases [25]. Peripheral blood eosinophil proportions of ≥3% in asthmatics suggest an inflammatory phenotype dominated by increased eosinophils, which can be used to predict the effectiveness of hormonal anti-inflammatory therapy in asthma [26,27].

In the GINA guidelines, a better reduction in the risk of acute asthma exacerbations is observed in patients with peripheral blood eosinophils ≥ 260/μL treated with omalizumab for asthma [28,29]. In this study, there was a statistically significant (*p* < 0.05) decrease in blood eosinophil count and percentage after 16 weeks of treatment, compared to beforehand. The unique inhibitory effect of omalizumab on IgE can inhibit the inflammatory response at the root, and as an important cell involved in the allergic reaction, the decrease in eosinophils can reflect the effectiveness of omalizumab and the good control of asthma symptoms.

Among the parameters of asthma assessment, lung function is the most objective measure of disease severity and control, and FEV1 and PEF quantify the degree of airway obstruction [30]. After omalizumab treatment, responding patients’ FEV1 improved at 6 months compared to beforehand, and then remained stable for 2 years [31]. A randomized controlled study by Pillai et al. [32] showed a decrease in lung function in patients in the placebo-treated group and an improvement in lung function in patients in the omalizumab-treated group through 20 weeks of treatment with omalizumab. The present study looked at changes in FEV1 and PEF in patients after the addition of omalizumab treatment, and it showed a significant improvement in FEV1 in responding patients (*p* < 0.001) compared to beforehand, while the improvement in PEF was not significant. Similar to the above study, these results have demonstrated the good effect of omalizumab in controlling symptoms and improving lung function, which is particularly important for the long-term quality of life of asthma patients.

Most of the patients had a history of allergy or combined with other allergic diseases, mostly combined with allergic rhinitis (55.17%) and atopic dermatitis (27.59%), and their symptoms of other allergic diseases were improved after treatment with omalizumab. Epidemiological surveys [33] show that 59.5–69.9% of asthma patients in China have allergic rhinitis in combination, and that allergic rhinitis mostly precedes asthma, and is an independent risk factor for the development of asthma [34,35,36]. Allergic rhinitis and asthma interact with each other, and patients with a combination of these two conditions often have worse outcomes than those with only one of the conditions, and so clinicians recommend management based on the principle of "one airway, one disease" and combined treatment [13,37]. Anti-IgE therapy has been shown to be effective in reducing the number of acute asthma attacks, reducing rhinitis symptoms, and improving the quality of life in patients with asthma combined with allergic rhinitis. The Chinese guidelines also recommend anti-IgE therapy for patients with allergic rhinitis combined with asthma that is clearly caused by IgE if the best conventional treatment and allergen avoidance have not been successful [38].

There are no standardized criteria for assessing the efficacy of biologically targeted asthma treatments. Currently, subjective criteria (e.g., ACT, Global Evaluation of Treatment Effectiveness Scale (GETE), the Asthma Control Questionnaire (ACQ), the Asthma Quality of Life Questionnaire (AQLQ), etc., and objective criteria (e.g., ICS dosage, OCS dosage, lung-function tests, number of acute exacerbations, etc.) are used to assess efficacy. In this study, OCS dosage, the number of acute exacerbations, and changes in ACT scores, as well as pulmonary function FEV1 and PEF, were used to assess efficacy. The ACT score was used as the evaluation criterion in this study, and patients with an improvement in ACT score up to a MID score of 3 after treatment [39], or with a poor or very poor level of control before treatment (ACT score 5–19) were defined as responding after treatment with good asthma control (ACT score 20–25). This definition reflects the improved level of asthma control, and correlation analysis has shown that it also better reflects the reduction in OCS use and the number of acute exacerbations, which have been used in some previous studies [15,39]. The GETE is a five-point scale of excellent (complete control of asthma), good (significant improvement), moderate (discernible but limited improvement), poor (no significant change), and worse, with a score of "excellent" or "good" often being used to evaluate the efficacies of biologically targeted asthma treatments. A score of “excellent” or “good” is often defined as being responsive to treatment [40,41]. This study found that after 16 weeks of treatment with omalizumab, patients had improved ACT scores, improved asthma and allergy symptoms, reduced OCS use, and fewer acute exacerbations, again validating the effectiveness of omalizumab in the Chinese population. However, the above study did not include a control group and was only compared with the baseline, which does not provide a good indication that the improvement in asthma control was due to the use of omalizumab and was not due to other treatments or other factors. In addition, the study did not include patients who had discontinued treatment before 16 weeks, which does not reflect that the short-term subcutaneous use of omalizumab did not benefit asthma patients, and further large cohort studies are needed to confirm this.

The response rate after 12 months of treatment with omalizumab was 64.7% in a large foreign study of 788 patients [39], as evaluated by an improvement in ACT scores, and the results of this study were similar, with a response rate of 68.75% after 16 weeks of treatment with omalizumab which is similar to the results of real studies conducted abroad in recent years (a 58.29–67.3% response rate) [42,43,44]. The effectiveness of omalizumab in the treatment of moderate-to-severe allergic asthma has been demonstrated in a number of large randomized controlled trials [45,46,47]. 

This study has the following shortcomings: it is a retrospective observational study, the sample size is small and needs to be validated in a multicenter prospective clinical study with a large sample, the observation period of 16 weeks may be too short and it cannot be excluded that the number of responders increases with the duration of treatment, and this study only analyzes the predictive effect of baseline total IgE levels and eosinophil counts on the response to omalizumab for asthma, and does not address dynamic changes; future repeated measures models could be performed for further study.

## 5. Conclusions

Omalizumab combined with ICS-LABA was effective in treating moderate-to-severe asthma in Chinese adults, with reduced oral hormone use, improved ACT scores, and fewer acute exacerbations after 16 weeks of treatment. Omalizumab can therefore be used as an add-on treatment. In addition, high serum IgE levels and the presence of comorbidities were predictors of the response to treatment. Moderate-to-severe allergic asthma poses a serious disease burden for Chinese patients, and the application of omalizumab in the clinic in 2018 brings new options for patients, but the number of patients currently using it is small due to geographical and economic reasons, and there is an urgent need to accumulate clinical application experience, research data, and more precise positioning to benefit more patients.

## Figures and Tables

**Figure 1 vaccines-10-02068-f001:**
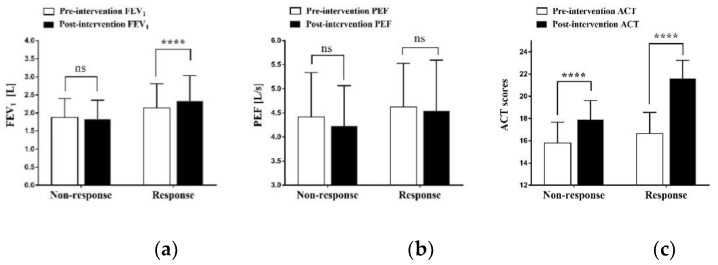
(**a**) Change in FEV_1_ after 16 weeks of treatment in non-response and response groups; (**b**) change in PEF after 16 weeks of treatment in non-response and response groups; (**c**) change in ACT score after 16 weeks of treatment in non-response and response groups; ns, no significance; ****, *p* < 0.001, a statistically significant difference.

**Figure 2 vaccines-10-02068-f002:**
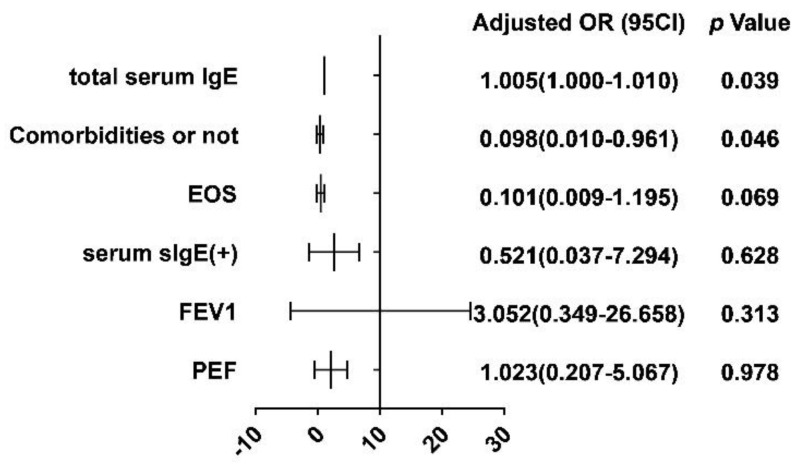
Binary logistic regression analysis of factors influencing omalizumab response.

**Table 1 vaccines-10-02068-t001:** Subcutaneous omalizumab doses for patients 12 years of age and older with asthma.

Baseline IgE (IU/mL)	Weight (kg)									
	21−25	26−30	31−40	41−50	51−60	61−70	71−80	81−90	91−125	126−150
31−100	75	75	75	150	150	150	150	150	300	300
101−200	150	150	150	300	300	300	300	300	450	600
201−300	150	150	225	300	300	450	450	450	600	375
301−400	225	225	300	450	450	450	600	600	450	525
401−500	225	300	450	450	600	600	375	375	525	600
501−600	300	300	450	600	600	375	450	450	600	
601−700	300	225	450	600	375	450	450	525		
701−800	225	225	300	375	450	450	525	600		
801−900	225	225	300	375	450	525	600			
901−1000	225	300	375	450	525	600				
1001−1100	225	300	375	450	600					
1101−1200	300	300	450	525	600			DO NOT DOSE		
1201−1300	300	375	450	525						
1301−1500	300	375	525	600						

Notes: 
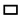
 Subcutaneous doses to be administered every 4 weeks; 
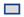
 Subcutaneous doses to be administered every 2 weeks; 
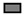
 Insufficient data to recommend a dose.

**Table 2 vaccines-10-02068-t002:** Comparison of baseline characteristics ^1^.

Characteristic	Total (n = 32)	Non-Response (n = 10)	Response (n = 22)	*P* ^3^
Gender				0.636
Male	13 (40.63%)	4 (40.00%)	9 (40.90%)	
Female	19 (59.37%)	6 (60.00%)	13 (59.10%)	
Age (years)	53.38 ± 13.61	57.90 ± 9.83	51.32 ± 14.77	0.205
Weight	62.11 ± 9.55	62.50 ± 9.62	61.94 ± 9.74	0.877
BMI ^2^				
Total	23.44 ± 2.91	23.73 ± 2.55	23.31 ± 3.11	0.705
Above normal	26.24 ± 1.61	26.52 ± 0.70	26.11 ± 1.91	0.670
Course (years)	4.00 (2.00–10.00)	4.50 (2.75–12.50)	3.00 (2.00–11.00)	0.580
Comorbidities				
RS	2 (6.25%)	1 (10.00%)	1 (4.55%)	
ACO	6 (18.75%)	5 (50.00%)	1 (4.55%)	
GERD	1 (3.13%)	1 (10.00%)	0 (0.00%)	
Allergy history				
Food	3 (9.38%)	0 (0.00%)	3 (13.64%)	
Inhalation	6 (18.75%)	1 (10.00%)	5 (22.73%)	
Drug	7 (21.87%)	3 (30.00%)	4 (18.18%)	
Skin	3 (9.38%)	1 (10.00%)	2 (9.09%)	
Taking OCS	11 (34.38%)	5 (50.00%)	6 (27.27%)	0.077
OCS dose	0.00 (0.00–13.75)	17.50 (0.00–22.50)	0.00 (0.00–2.50)	0.024
Total serum IgE	503.80 (315.68–956.00)	410.03 ± 304.63	630.25 (374.30–1035.50)	0.016
EOS	0.43 (0.10–0.78)	0.74 ± 0.47	0.39 (0.08–0.49)	0.064
EOS%	5.20 (1.63–8.40)	4.05 (1.73–14.18)	5.20 (1.50–7.55)	0.745
FEV1	2.06 ± 0.63	1.88 ± 0.52	2.14 ± 0.67	0.285
PEF	4.56 ± 0.90	4.42 ± 0.92	4.62 ± 0.91	0.562
ACT scores	16.38 ± 1.91	15.80 ± 1.87	16.64 ± 1.92	0.258
Exacerbations	0.00 (0.00–1.00)	2.00 (1.00–2.50)	0.00 (0.00–1.00)	<0.001

Notes: BMI, body mass index; N, number of cases; RS, rhinosinusitis; ACO, asthma-chronic obstructive pulmonary disease overlap; GERD, gastroesophageal reflux disease; OCS, oral corticosteroids; EOS, blood eosinophils; FEV1, forced expiratory volume in 1 s; PEF, peak expiratory flow; ACT, asthma control test. ^1^ Result indicates: mean ± SD, median (IQR), and number of cases (percentage). ^2^ Normal: <24 kg/m^2^ (China standard, adjusted according to the World Health Organization). ^3^ Independent samples *t*-test (conforming to a normal distribution) or Mann-Whitney U-test (not conforming to a normal distribution).

**Table 3 vaccines-10-02068-t003:** Differences before and after treatment ^1^.

	Before	After	Z ^2^ or *t* ^3^	*P*
Taking OCS	11/32 (34.38%)	7/32 (21.88%)	12.250	<0.001
OCS dose	0.00 (0.00−13.75)	0.00 (0.00−20.00)	−2.986	0.003
Total serum IgE	503.80 (315.68−956.00)	299.75 (168.13–571.68)	−3.871	<0.001
EOS	0.43 (0.10−0.78)	0.18 (0.07–0.76)	−2.974	0.003
EOS%	5.20 (1.63–8.40)	2.10 (1.55−4.80)	−2.488	0.013
FEV1	2.06 ± 0.63	2.17 ± 0.69	−3.712	0.001
PEF	4.56 ± 0.90	4.44 ± 0.99	0.855	0.399
ACT scores	16.38 ± 0.34	20.44 ± 0.42	−11.315	<0.001
Exacerbations	0.00 (0.00−1.00)	0.00 (0.00−1.00)	−3.051	0.002

Note: OCS, oral corticosteroids; EOS, blood eosinophils; FEV1, forced expiratory volume in 1 s; PEF, peak expiratory flow; ACT, asthma control test. ^1^ Result indicates: mean ± SD, median (IQR), and number of cases (percentage). ^2^ Two correlated samples Wilcoxon test. ^3^ Paired-sample *t*-test.

**Table 4 vaccines-10-02068-t004:** Differences in different responses ^1^.

	Non-Response (n = 10)				Response (n = 22)			
	Before	After	Z ^2^ or *t* ^3^	*P*	Before	After	Z ^2^ or *t* ^3^	*P*
Taking OCS	6/10 (60.00%)	5/10 (50.00%)	0.200	0.655	5/22 (22.73%)	2/22 (9.09%)	20.455	<0.001
OCS dose	17.50 (0.00−22.50)	5.00 (0.00−14.38)	−2.260	0.026	0.00 (0.00−2.50)	0.00 (0.00−0.00)	−2.060	0.039
Total serum IgE	410.03 ± 304.63	465.28 ± 301.78	−1.165	0.274	630.25 (374.30−1035.50)	272.10 (149.90–469.65)	−4.107	<0.001
EOS	0.74 ± 0.47	0.77 ± 0.57	−0.169	0.869	0.39 (0.08−0.49)	0.13 (0.05−0.29)	−3.247	0.001
EOS%	4.05 (1.73−14.18)	3.15 (1.88−12.35)	−0.714	0.475	5.20 (1.50−7.55)	1.85 (1.38−3.08)	−2.420	0.016
Exacerbations	2.00 (1.00−2.50)	1.00 (1.00−1.25)	−2.121	0.034	1.00 (1.00−1.25)	0.00 (0.00−0.00)	−2.236	0.025

Note: OCS, oral corticosteroids; EOS, blood eosinophils; ^1^ Result indicates: mean ± SD, median (IQR), and number of cases (percentage). ^2^ Two correlated samples Wilcoxon test. ^3^ Paired-sample *t*-test.

## Data Availability

Data are available, upon reasonable request, by emailing: djj910625@163.com.

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
