# Peer review of "Response to Omalizumab as an Add-On Therapy in the Treatment of Allergic Asthma in Adult Chinese Patients—A Retrospective Study"

_vaccines, 2022, doi:10.3390/vaccines10122068_

Round 1
Reviewer 1 Report
I carefully revised the manuscript entitled “Response to budesonide formoterol and omalizumab combination therapy: a retrospective study in adults with allergic asthma” by Li et al.
This paper describes a study that ailed to evaluate the efficacy of the combination of the Omalizumab with the budesonide formoterol in the treatment of the allergic asthma. The study is interesting and seems well conducted, and the results give some new knowledge that can help in the approach to this pathological condition. However, before to be considered suitable for publication, the manuscript needs some major revisions.
First, the authors should better explain some points of their protocol as well as some choices:
- In line 111, the authors stated that the clinical data were collected retrospectively, while in the rest of the manuscript there is no mention about the retrospective character of their study. Please, if this is a retrospective studies, in which the parameters were chosen after and not before, thus with some limitations in the range of choice, this point should be clearly stated.
- Lines 156-163: the rational at the base of dosage pf the omalizumab is explained, however a clear resume (maybe with a table) of the dose used for each patient could help the reader to better understand.
- Since a real control group is lacking (for example a group of patients treated with a well-known therapy), the simple comparison between before and after mean values could alter the real interpretation of the efficacy. I suggest to calculate the delta values for each patient, and after to use these data to obtain the mean values for each parameter. The delta values help to better represent the change between before and after the treatment (improvement or worsening) normalizing the eventual great changes of some patients, while the simple difference between means is less precise.
- Lines 126-129, always regarding the way you used to evaluate the improvement before/after: why did you chose to use the MID in these two manners to obtain groups? Have you some previous bibliography to cite to support your choices? If not, can you explain why did you chose this approach?
- Taken together, all these observations show the last paragraph of the introduction should be partially rewrite (or also the first part of the M&M) in order to better present the design and the choices at the base of your protocol.
Second, the manuscript presents some inaccuracies that should be amended:
- Abstract: the implication of the budesonide formoterol in combination with the omalizumab is mentioned only once in the end of the abstract. Please modify the text (also in the manuscript) to make clear that the study evaluate only the combination omalizumab + budesonide formoterol.
- Lines 132-133: this sentence is a complete tautology, please delete it.
- Lines 164-168: you repeated twice the distribution between males and females.
- Lines 170-172, and table 1 : it would be useful to clarify the drugs, the foods and the inhalants the patients were allergic to.
- Table 1: change DRED into GRED.
- Table 1 vs Table2: the SD of the global ACT score before the treatment are different between table 1 and table 2. This is not possible.
- Table 2: it is not clear to me why the values of the OCS dose and exacerbations (before and after) are always 0.00 and why if they did not change the p-value is significant.
- Discussion: again, you began talking about omalizumab effect, while the study evaluated (and confirmed) the omalizumab + budesonide formoterol effect.
- Discussion: it would be very interesting to compare the improvements obtained in you study with data reported in bibliography for other treatments. In the current form, it is not possible to understand if the combination you tested can guarantee better results than other therapies.
Reviewer 2 Report
Li N et al., presented retrospective data regarding the effect of Omalizumab in allergic asthmatic patients treated with ICS/LABA (budesonide/formoterol). The first aim is the efficacy of omalizumab in combination with budesonide/formoterol in the treatment of moderate to severe allergic asthma. The second aim is to investigate whether baseline clinical characteristics such as total serum IgE and eosinophil levels are predictive of response to treatment. Results agreed with previous published data without any particular novelty. The number of analyzed subjects is low and the follow-up short to properly evaluate disease exacerbations.
COMMENTS:
Please indicate in the abstract the characteristics of enrolled patients instead of patients who fulfilled inclusion criteria.
Line 32: please correct the sentence: It has different clinical phenotypes
Lines 95-96 Please eliminate the description of the reference 13
Describe in the method section the evaluation of free IgE
Line 175 please substitute ratio with percentage
At the beginning of paragraph 3.2 authors should avoid to repeat criteria for definition of response and non response group already indicated in the method section
Please change or explain phrase at lines 289-291
Authors reported in the discussion that most of the patients with allergic comorbidities improved the related symptoms after Omalizumab treatment, however, they did not report these data in the result section
Reviewer 3 Report
I have reviewed the manuscript with an ID Vaccines-2007954 entitled Response to budesonide formoterol and omalizumab combination therapy a retrospective study in adults with allergic asthma for Vaccines.
The authors analyzed the efficacy of omalizumab in combination with budesonide/formoterol in the treatment of moderate to severe allergic asthma in a cohort of Chinese patients. The study requires major revisions before it should be considered for publication.
1. I suggest changing the title to correspond better to the article content and study objective. I propose „Response to omalizumab as add-on therapy in the treatment of allergic asthma in adults Chinese patients - a retrospective study”
2. In the abstract - abbreviation should be explained at the first appearance in the text.
3. The aim of the study is not clearly established. The author described the method instead of the objective. I suggest to state clearly what was the aim of the study.
4. Why patients with moderate asthma were included in the present study and why did they receive omalizumab as add-on therapy? GINA guidelines recommend omalizumab as add-on therapy in step 5 (severe asthma).
5. The authors mentioned as inclusion criteria - poor control after ≥3 months of conventional budesonide/formoterol therapy. They should mention the exact dose for budesonide. What was the maximal accepted dose for budesonide to consider lack or poor response?
6. Results section – lines 164-167 – a duplication of the same phrase
7. The authors mentioned 6 cases of COPD. This means that 6 patients had Asthma-COPD overlap.
8. The authors should also explain what that means allergic to skin. The patients had associated chronic urticaria and atopic dermatitis.
9. In table 1 the authors should present the allergens (respiratory allergens, foods, or drugs) or the clinical manifestations of the allergies (allergic rhinitis, anaphylaxis) not a mixture of them, in the allergy history section.
10. Information from lines 193-195 are presented previously in material and method when the authors defined the criteria of asthma response. They should exclude it from the results section.
11. Percentage of Eo should be excluded. The absolute values are used in defining asthma phenotype and monitoring treatment response.
12. In Line 208 the authors mentioned a "significant reduction in ACT scores” after omalizumab therapy. I think they refer to the improvement of ACT score after 16 weeks of treatment (in table 2 there is an increase in ACT score after treatment). Please revise the obtained data.
13. The authors should decide how they present the results (table of the figure) in order to avoid results duplication. The same results are presented in table 3 and figure 1.
14. Revise the name of the monoclonal antibody – line 260.
15. Line 266 – what means anti-omalizumab?
16. Why did they comment in the discussion the GETE score, since the authors did not use it?
17. Lines 344-346 -.The effectiveness of omalizumab in the treatment of moderate to severe allergic asthma has been demonstrated in a number of large randomized controlled trials[45] - but the author included a single reference.
18. Exclude the last phrase of the discussion. Not important for the present study. The authors did not investigate the mechanism of omalizumab.
19. Limitations of the study should be included in the discussion section
20. Check reference 43. It is not properly cited
Round 2
Reviewer 1 Report
I’ve carefully reviewed the revised version of this manuscript.
Unfortunately, even if I can appreciate the efforts that the authors tried to do to improve the manuscript answering to reviewers’ comments, I don’t find the text improved enough to be considered it suitable for publication. Several points and doubts that I’ve raised during the first review did not lead to significant changes of the manuscript. The authors preferred to keep their points of view, which is comprehensible, but, in my opinion, it does not allow an improvement of the paper. Moreover, I must agree with the other reviewer that pointed out the problem of the small population studied.
Reviewer 2 Report
Please correct GRED with GERD in table 2
Author Response
Please see the attchment.

Reviewer 3 Report
I have reviewed the revised manuscript with an ID Vaccines-2007954 entitled Response to budesonide formoterol and omalizumab combination therapy a retrospective study in adults with allergic asthma for Vaccines.
The authors analyzed the efficacy of omalizumab in combination with budesonide/formoterol in the treatment of moderate to severe allergic asthma in a cohort of Chinese patients. The authors slightly improved the manuscript based on the reviewers' opinions. The following queries remained without changes and answers. I suggest taking into consideration these recommendations in order to improve the mansucript and make it easily readable.
1. The aim of the study is not clearly presented and is in the same paragraph as in the oldest version.
2. The authors mentioned as inclusion criteria - poor control after ≥3 months of conventional budesonide/formoterol therapy. They should mention the exact dose for budesonide (160/4.5 mcg twice daily or 320/9 mcg twice daily). What was the maximal accepted dose for budesonide to consider lack or poor response? It is not enough to mention 1 or 2 inhalations if the concentration is not mentioned.
3. The authors mentioned 6 cases of Asthma-COPD overlap. I suggest writing ACO instead of COPD in table 2.
4. In the same table 2 GRED means GERD?
5. The authors mentioned in the answer letter that the patients had associated chronic urticaria, and atopic dermatitis but they don’t change in table 2.
6. In table 1 the authors should present the allergens (respiratory allergens, foods or drugs) or the clinical manifestations of the allergies (allergic rhinitis, anaphylaxis) not a mixture of them, in the allergy history section.
7. Presentation of the same results in both table and figure means to duplicate the results. I suggest maintaining the figures and the presented parameters in figure 1 should be excluded from table 4.
8. If GETE score is not used in the evaluation of asthma control, I suggest excluding the corresponding part from the discussion.
Round 3
Reviewer 3 Report
The article has improved significantly. Some changes are still required.
1. The aim of the study is still the old one (please revise it).
2. You should exclude completely the information regarding GETE score (line 353-355).
The reviewer